# Mapping Maximum Tree Height of the Great Khingan Mountain, Inner Mongolia Using the Allometric Scaling and Resource Limitations Model

**Yao Zhang [1], Yuli Shi [1,\*], Sungho Choi [2]** **, Xiliang Ni [3] and Ranga B. Myneni [2]**

[1]   School of Remote Sensing and Geomatics Engineering, Nanjing University of Information Science
      and Technology, Nanjing 210044, China; yao_zhang_ll@outlook.com
[2]   Department of Earth and Environment, Boston University, Boston, MA 02215, USA; schoi@bu.edu (S.C.);
      rmyneni@bu.edu (R.B.M.)
[3]   State Key Laboratory of Remote Sensing Science, Institute of Remote Sensing and Digital Earth,
      Chinese Academy of Science, Beijing 100101, China; nixl@irsa.ac.cn
\*   Correspondence: ylshi.nuist@gmail.com; Tel.: +1-585-184-8199

**Abstract:** Maximum tree height is an important indicator of forest vegetation in understanding the properties of plant communities. In this paper, we estimated regional maximum tree heights across the forest of the Great Khingan Mountain in Inner Mongolia with the allometric scaling and resource limitations model. The model integrates metabolic scaling theory and the water–energy balance equation (Penman–Monteith equation) to predict maximum tree height constrained by local resource availability. Monthly climate data, including precipitation, wind speed, vapor pressure, air temperature, and solar radiation are inputs of this model. Ground measurements, such as tree heights, diameters at breast height, and crown heights, have been used to compute the parameters of the model. In addition, Geoscience Laser Altimeter System (GLAS) data is used to verify the results of model prediction. We found that the prediction of regional maximum tree heights is highly correlated with the GLAS tree heights ($R^2$ = 0.64, RMSE = 2.87 m, MPSE = 12.45%). All trees are between 10 to 40 m in height, and trees in the north are taller than those in the south of the region of research. Furthermore, we analyzed the sensitivity of the input variables and found the model predictions are most sensitive to air temperature and vapor pressure.

**Keywords:** maximum forest heights; metabolic scale theory; allometric scaling and resource limitation model

## 1. Introduction

   Forests, as a crucial part of terrestrial vegetation, play a central role in regulating the carbon and water cycles [1–4]. Moreover, height is an important indicator of various forest features, such as biological productivity, mortality rates, canopy density, and energy exchange [5–9].

   Several articles have reported nonphysical or nonphysiological approaches to generate spatially continuous maps of forest heights by combining remote sensing data and in situ measurements. It is possible to estimate tree height with optical data and altimeter data from terrestrial, airborne [10,11], and spaceborne LiDAR [12–16]. Airborne LiDAR and stereo-photogrammetry data can effectively reflect the vertical structure of forests, but its application is constrained to small regional scale due to the expensive costs [17]. While the spaceborne LiDAR can provide global elevation information, the sampling density is insufficient for the complete monitoring of equatorial and midlatitude forests [18]. In addition, the underlying physical and biological principles of forest growth are

often ignored in those approaches, and such neglect may lead to nonmechanistic shifts in the modelled outputs that are easily affected by the quality and quantity of training data [19].

Recent studies have applied spatial statistics and biophysical theories to establish biophysical models [20–23]. The model can give spatially continuous canopy heights of forests at large scale with the sparse observations and geospatial predictors like climatic variables and topography [20]. Climatic variables are good candidates for predictors of such models based on an assumption that climate regulates overall plant growth [24–26]. The model we used here, called allometric scaling and resource limitations (ASRL), is a biophysical model. The ASRL model integrates metabolic scaling theory (MST) for plants [27] and the water–energy balance equation [28] to predict potential maximum tree heights [29,30]. In ASRL model, the biophysical principles provide a generalized mechanistic understanding of relationships between tree size and geospatial parameters, including topography and climatic variables [29]. This model can be used for monitoring forests at large scales.

However, the original model is not suitable for some study areas due to differences in forest growth status, such as canopy density, stand age, and stand density [20,31]. In order to solve this problem, the ASRL model was improved in this paper to be highly consistent with the forest growth status in the study area. The improved ASRL model was used to map continuous maximum forest canopy heights of the Greater Khingan Mountain in Inner Mongolia with actual measurements, climatic data, and remote sensing data.

## 2. Data

The study area is situated in the Great Khingan Mountain, located within cold temperate continental monsoon climate zone of northeast Inner Mongolia, China (119°36′–125°24′ E, 47°03′–53°20′ N). It is hot and humid in summer, but cold and dry in winter. The annual average temperature is about −3.5 °C, while extreme low temperature can reach −50 °C. The annual mean precipitation in the study area is approximately 300–450 mm. The main forest in the study area is a mix of *Larix gmelinii* and White birch, which is formed by White birch's invasion after the destruction of the native *Larix gmelinii* forest. The forest covers approximately 8.17 million ha, with an elevation range 250–1745 m above sea level.

Field measurement data were derived from the ground survey data in Genhe city in August 2013 and 2016. Ninety plots were established and measured, including 19 square plots (45 × 45 m, or 30 × 30 m) and 71 circular plots (radius = 10, or 15 m). Figure 1 presents the distribution of these plots. The centers of each plot were located using Trimble GeoHX6000 Handheld GPS (Trimble, Sunnyvale, CA, USA) with an accuracy of approximately 2–3 m. Within each plot, diameter at breast height (DBH) of all live trees were measured using a diameter tape but only DBH over 5 cm were recorded. Trupulse TM2000 was used to measure tree height and height to crown base for each stand tree. Crown widths were approximated to the average of two values measured along two perpendicular directions from the location of the tree top. In order to avoid double counting of trees, latest record was used if a tree was measured more than once.

For input climate data, including monthly precipitation, wind speed, vapor pressure, air temperature and solar radiation, we used the WorldClim Version 2.0 (Sustainable Intensification Innovation Lab, Kansas State University, Manhattan, KS, USA) dataset averaged over multiple years from 1970 to 2000 at a 1-km spatial resolution (http://worldclim.org/version2). The input elevation data were derived from ASTER Global Digital Elevation Map (GDEM) V2 at a 30-m spatial resolution. All input gridded data were resampled and reprojected at a 1-km spatial resolution with a Lambert Conformal Conic map project to generate the continuous map of tree heights.

Two types of Moderate Resolution Imaging Spectroradiometer (MODIS) products were used as ancillary data in this study. Vegetation classification based on IGBP [32] derived from MODIS land cover type product (MCD12Q1) at a 500-m spatial resolution was used to define forest area (Figure 2a). Another ancillary data named MODIS Vegetation Continuous Filed (VCF) at 250-m spatial resolution

was used to identify forest land with percent tree cover over 40 (Figure 2b). The ancillary data was at the same spatial resolution and had the same projection as the input data.

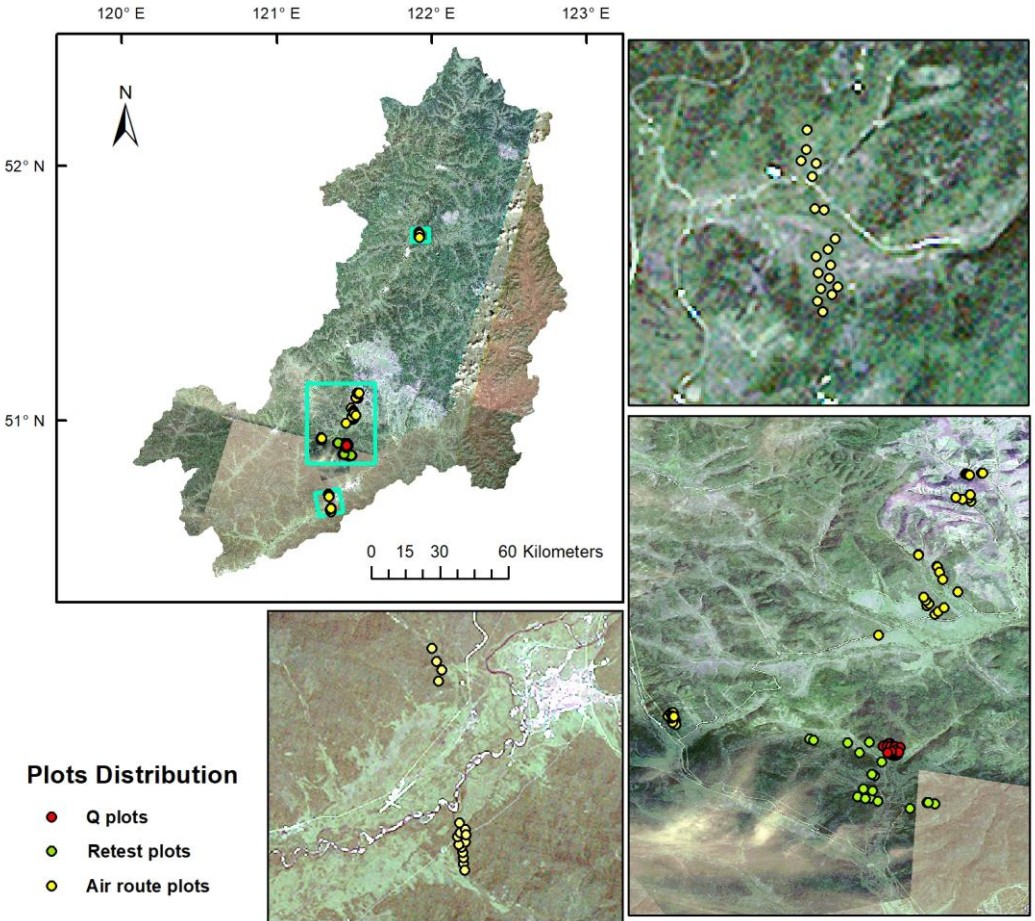

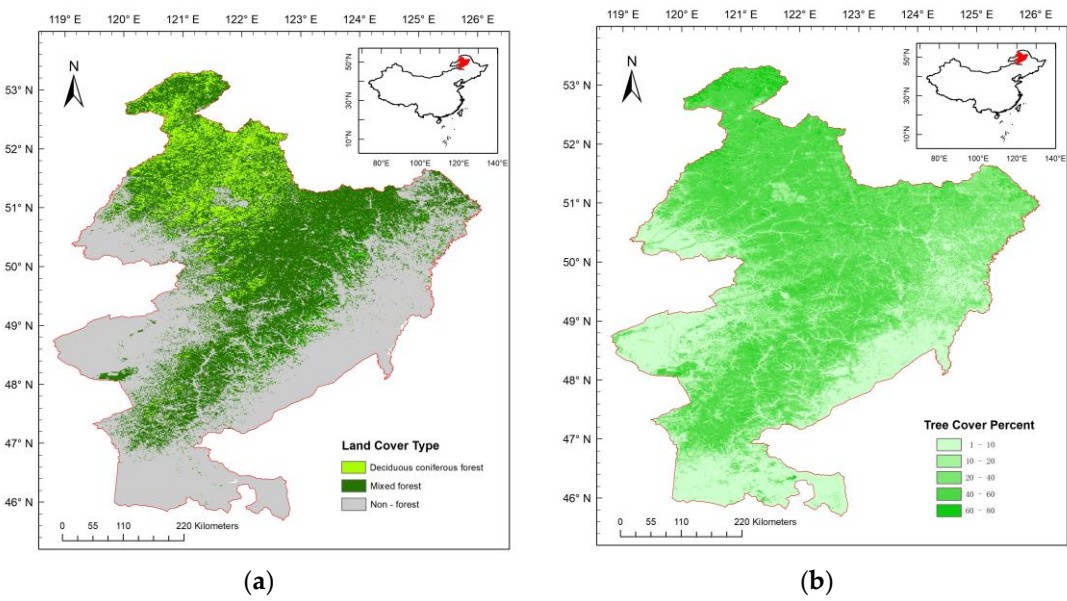

**Figure 1.** Distribution of plots in Genhe city.

**Figure 2.** (**a**) Distribution of five surface cover types of the Greater Khingan Mountain in Inner Mongolia in 2013; (**b**) Vegetation coverage rate of the Greater Khingan Mountain in Inner Mongolia in 2013.

Global Surface Altimetry Data (GLA14 product) from 2003 to 2005 was used to extract maximum tree heights to verify predictions of the ASRL model. The distribution of GLAS footprints is in Figure 3. According to Ni's [23] research, when slope is smaller than 10, GLA14 product performs highest accuracy in maximum tree height's extraction. The best equation to estimate forest heights is:

$$H = (W_{SB} - W_{GP}) - d * \frac{\tan \theta}{2} \tag{1}$$

where $W_{SB}$ represent the signal beginning and $W_{GP}$ is the ground peak of GLAS full- waveform. While $d$ is spot size and $\theta$ is topographic slope.

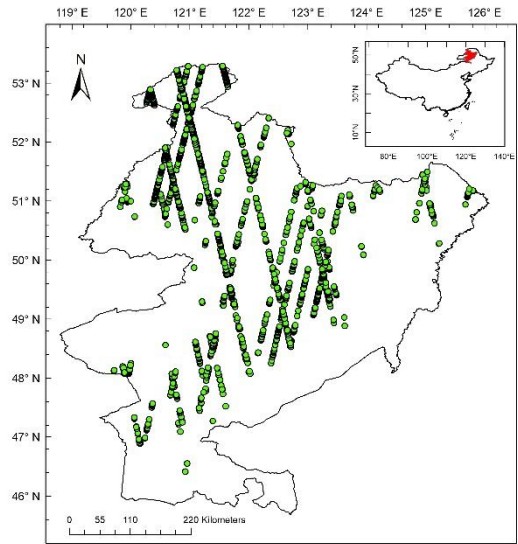

**Figure 3.** Distribution of GLAS footprints in the Great Khingan Mountain, Inner Mongolia.

## 3. Methods

### 3.1. The ASRL Model Framework

Biologists have found that the size and structure of living organisms have a great influence on their physiological processes [33,34]. In order to meet the needs of physiological processes, there is a stable proportional coefficient among the internal structure of the organism that accompanies its growth. The MST assumes that the plant metabolic rate B scales with the size of the whole plant, including volume V and mass M as: $B \propto V^\theta \propto M^\theta$ [35], and the parameter $\theta$ is close to 3/4. Kempes C.P. et al. [20] proposed ASRL tree height model which combines the metabolic scaling theory and energy balance equation. The ASRL model assumes that: (1) the tree can extract the resources from the environment which are needed for growth; (2) the ability of absorbing resources depends on the size of the tree; and (3) the resources that the environment can supply limit the growth of the tree. In the model, this is expressed by inequalities of three flow rates: $Q_0 \leq Q_e \leq Q_p$. The evaporation flow rate ($Q_e$) of a tree must satisfy its minimum metabolic flow rate ($Q_0$) but not exceed the potential rate of water inflow ($Q_p$) that the external environment can provide. These water flow rates are affected by both tree size and local environment supply. Based on the scale growth theory, we can use tree height to represent other characteristics of the tree, and the water and energy in the environment can be calculated by climatic predictors (such as temperature, pressure, water vapor pressure, solar radiation and precipitation). The basic framework of the model is shown in Figure 4.

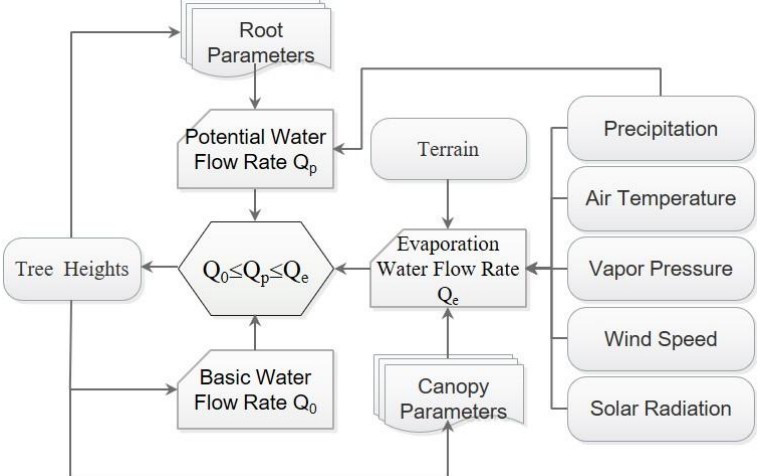

**Figure 4.** The basic framework of the allometric scaling and resource limitation (ASRL) model.

The basic water flow rate $Q_0$ is expressed as the equation of the tree height h:

$$Q_0 = \sum_{}^{12\text{months}} \beta_1 h^{\eta_1} \tag{2}$$

where $\beta_1$ and $\eta_1$ are the constant and exponent for basal metabolism. The potential water flow rate $Q_p$ is based on tree height h, elevation, and precipitation:

$$Q_p = \sum_{}^{12months} \gamma\left(2\pi r_{root}^2\right)\Psi P_{inc} \tag{3}$$

The root absorption efficiency $\gamma$ is related to local soil properties and terrain, and the $2\pi r_{root}^2$ is hemispheric root surface area [36,37]. The normalized terrain index $\Psi$ is calculated from the elevation data, and $P_{inc}$ is the input precipitation data. Evaporation water flow rate is given as a function of tree height h and climatic variables, including wind speed, solar radiation, temperature, precipitation, and vapor pressure:

$$Q_e = a_L v_{water} \sum_{}^{12months} E_{flux} \tag{4}$$

The effective tree area $a_L$ is calculated from the single leaf area $s_{leaf}$ and the branching architecture [20]. The molar volume of water $v_{water}$ can be calculated from the molar mass of water and the water density. The evaporative molar flux $E_{flux}$ is related to climatic factors such as temperature, water vapor pressure, and wind speed.

### 3.2. Improvements in the ASRL Model

Previous studies have found reasons for deviations from basic MST, including tree species, plant interaction, self-competition, and forest age [38,39]. The correlation established in the original model is difficult to reproduce in different research areas or times. According to Choi's [29] research, this paper makes the following improvements to the ASRL model to adapt to the research area. Key parameters in the ASRL model are presented in Table 1.

First, the growth coefficient of tree height and DBH in the MST model ($h \propto r_{stem}^{\phi}, \phi \approx 2/3$) is replaced with a statistic value of 0.7153. The measured tree height and DBH data is used to reconstruct the forest allometric scaling equation of the Greater Khingan Mountain in Inner Mongolia and replace the theoretical value of $\phi$ in the MST model. It reflects the differences in forest metabolism and metabolic variability in different regions [38,39].

Second, we replaced the scale factor of tree height $h$ and crown height $h_{cro}$ in the MST model ($h_{cro} \approx 0.79h$) with a statistic value of 0.47. Trees need to change their crown geometries and metabolic properties for the interplant interactions and self-competition [40,41]. The relationship between tree height and crown height in MST is unreliable, especially in the virgin forests of complex growth. The measured tree height and crown height data is used to reconstruct the forest allometric scaling equation of the Greater Khingan Mountain in Inner Mongolia and replace the theoretical value of 0.79 in the MST model.

Third, a dimensionless normalized topographic index $\Psi$ is introduced to reflect local terrain features. Generally, the flow of water always flows from high to low, and the terrain will inevitably affect the collection of water flow. In this paper, we introduced a dimensionless topographic index to simulate the situation:

$$\Psi = \ln[CA / \tan(slp)] / ln[CA_0 / \tan(slp_0)] \tag{5}$$

where $CA$ is catchment area and $slp$ is terrain slope. Assuming that the catchment area at hill top: $CA_0$ is 1, and slope at flat: $slp_0$ is $e^{-10}$. The topographic index of each pixel is calculated with DEM data, indicating the collection of precipitation due to effect of slopes.

Fourth, the canopy is treated as a huge leaf, and the energy exchange of the whole-plant is calculated based on the PM equation [28]. The soil heat flux G is also added into the energy balance:

$$R_{abs} = L + G + H + \lambda E_{flux} \tag{6}$$

where, the $R_{abs}$ is absorbed solar radiation, $L$ is thermal heat, and $H$ is sensible heat.

Fifth, based on the measured tree height data, $\beta_1$, $\gamma$, and $s_{leaf}$ are optimized. In the ASRL model, $\beta_1$ is metabolic coefficient of a tree, and its theoretical value is 0.017, which is determined by the biological mechanism of a tree. $\gamma$ is water absorption rate of roots with a theoretical value of 0.5. The value of water absorption rate may change in some soil types and environments. $s_{leaf}$ is the area of a single-leaf with a theoretical value of 0.001. Accompanying the tree's growth, the single leaf area will gradually change. These three parameters can't be obtained by direct measurement or calculation, but are important to the model: the basic water flow rate $Q_0$ is determined by $\beta_1$, while the value of $\gamma$ can affect the potential water flow rate $Q_p$, and the size of $s_{leaf}$ can determine the water and energy metabolism rate of the whole tree. In order to obtain these three parameters, a nonlinear multivariate optimization equation is constructed:

$$D\{\beta_1, \gamma, S_{leaf}\} = \sum_n \left\{ \left[ h_{obs} - h_c(\beta_1, \gamma, S_{leaf}) \right]^2 \right\} \tag{7}$$

where, $h_{obs}$ is measured tree height and $h_c$ is the modeled tree height. By iteration, when the D value is the minimum, the parameters are considered optimal.

**Table 1.** Key parameters in the ASRL model compared with previous studies.

| Parameters | Description | Initial values | Optimized Values |
|:---:|:---:|:---:|:---:|
| $\Phi$ | Exponent for tree height and stem radius allometry | 2/3 | 0.7153 |
| $\beta_1$ | Normalization constant for the basal metabolism | 0.0177 | 0.005 |
| $\gamma$ | Water absorption efficiency | 0.5 | 0.31 |
| $\Psi$ | Topographic index | \ | Calculated by slope and catchment area |
| $\beta_3$ | Crown ratio | 0.79 | 0.47 |
| $s_{leaf}$ | Area of single leaf | 0.001 | 0.0004 |

## 4. Results

Based on the improved ASRL tree height model, we generated the map of maximum tree heights of the Great Khingan Mountain in Inner Mongolia (Figure 5a). Tree heights in the research area are not more than 40 m. Trees in the north are taller than those in the south. Modelling tree heights are verified with the GLAS tree heights in the research area, and the results are shown in Figure 5b–d. The maximum tree height in ASRL predictions has a statistically significant linear relationship with the GLAS height ($R^2$ = 0.64, RMSE = 2.87 m, PMSE = 12.45%).

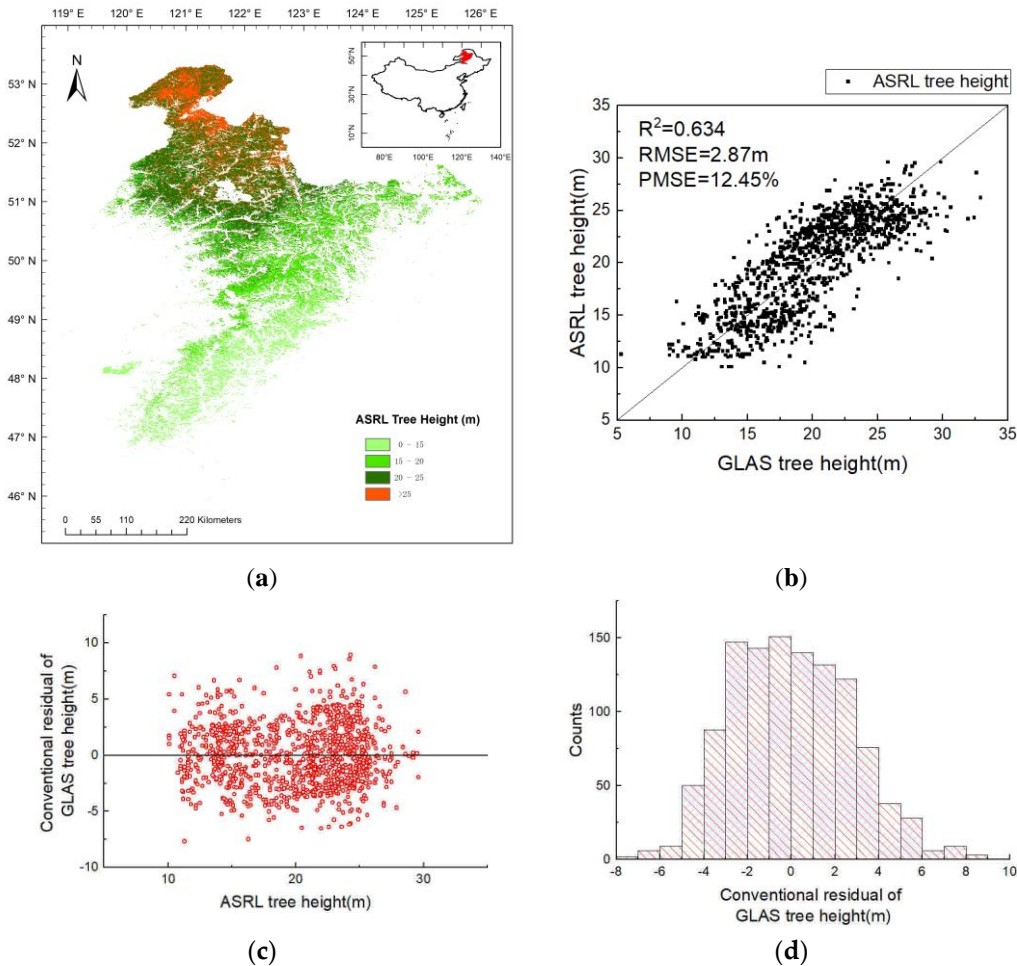

**Figure 5.** Inversion and verification results of the ASRL tree height model. (**a**) The distribution map of the maximum tree heights of the Great Khingan Mountain in Inner Mongolia based on the predictions of improved ASRL model (unit: m). (**b–d**) Three kinds of verification results: (**b**) The linear fitting of ASRL tree heights and GLAS tree heights ($R^2$ = 0.634, RMSE = 2.87m, PMSE = 12.45%); (**c**) The residual distribution of ASRL tree heights to GLAS tree heights, and (**d**) the counts of (**c**).

## 5. Discussions

### 5.1. Model Improvement

Kempes' model is based on metabolic scaling theory and resource constraint theory, and theoretical values of model parameters are given and applied to tree height calculations. In real-world applications of this model, these parameters have to be replaced and optimized to make the tree height model highly consistent with the forest growth status in the study area. The optimization of the model includes three items: parameter replacement, parameter optimization, and introduction of new parameters. The point with coordinates 121.554° E and 53.291° N is selected as the verification point to verify the

results of each optimization by controlling variables. The measured tree height of the verification point is 24.6 m. Climatic data of the verification point are imported into the model before and after optimization, and the inversion results are compared and analyzed. Therefore, this paper constructs the ASRL model in four cases: no parameters replacement (NPR), no parameters optimization (NPO), no topographic index (NTI), and the optimized model (OM).

In the original ASRL tree height model, the tree height $h$ and DBH $r_{stem}$. meets the following rule: $h \propto r_{stem}^{\phi}, \phi \approx 2/3$. Enquist et al. and Kempes et al. found that crown height $h_{crow}$ and tree height $h$ are required to be: $h_{cro} \approx \beta_3 h, \beta_3 = 0.79$. In order to improve the fit degree of the model to the research area, this paper utilizes the field data of tree heights, DBH and crown height in the Genhe city to establish the growth relationship between DBH, crown height and tree height, respectively. The results are shown in Figure 6. According to the measured data, parameter $\phi$ was 0.715, and the growth coefficient of crown height and tree height is $h_{cro} \approx 0.47h$.

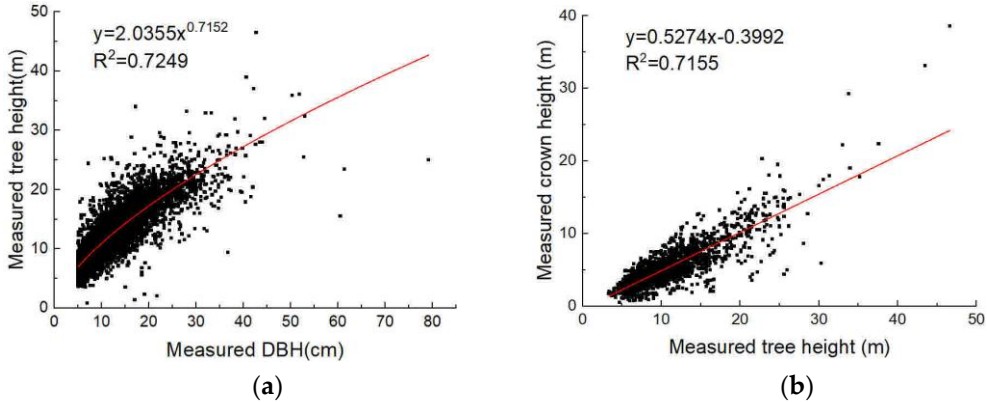

**Figure 6.** Modeling results of measured data. (**a**) The linear fitting result of measured tree height and diameter at breast height (DBH) ($R^2$ = 0.7249, $\Phi$ = 0.7249). (**b**) The linear fitting result of measured tree height and crown height ($R^2$ = 0.7155, $\beta_3$ = 0.47).

A cost function $D$ solved by the constrained nonlinear multivariable optimization is used to optimize the three parameters: $\beta_1$, $\gamma$, and $S_{leaf}$. The initial ASRL parameters were $\beta_1 = 0.01$, $\gamma = 0.5$ and $s_{leaf} = 0.001$ [7,20]. Inputting maximum tree height of each measured field as a sample, we minimized the cost function $D$ by calibrating all three parameters within ranges ($0.005 < \beta_1 < 0.02$, $0.01 < \gamma < 1$ and $0.0001 < s_{leaf} < 0.01$). Finally, the optimal parameters obtained in this paper were $\beta_1 = 0.005, \gamma = 0.31, S_{leaf} = 0.0004$.

With no parameter changes, including NPR (Figure 7b) and NPO (Figure 7c), the ASRL predictions at verification point are smaller than actual measurement. Comparing the curves of three kinds of water flow rates, the basic water flow rate and the potential water flow rate are not affected, but the actual evaporation water flow rate is significantly increased, which leads the intersection point of $Q_p$ and $Q_e$ to shift left and the predicted tree height to be smaller. The prediction of model without the normalized topographic index $\Psi$ is 33.2 m. As Figure 7d shows, the potential water flow rate is clearly increased, which leads the intersection point of $Q_p$ and $Q_e$ to shift right and the inversion result to be higher. Comparing with the result of the optimized model (Figure 7a), which is 22.7 m, we found: (1) parameter adjustment can make the result of evaporation water flow rate more reasonable and it effectively avoids underestimation of high trees; (2) the introduction of normalized topography index can reduce the sink flow in the high-terrain area and increase the sink flow in the low-terrain area, so that the convergence of precipitation on the surface in the model is consistent with the actual situation and the prediction accuracy of tree height has been improved. In addition, the curves of the optimized model show that maximum potential tree heights are mainly limited by water supply, meaning the verification point is a water-limited environment.

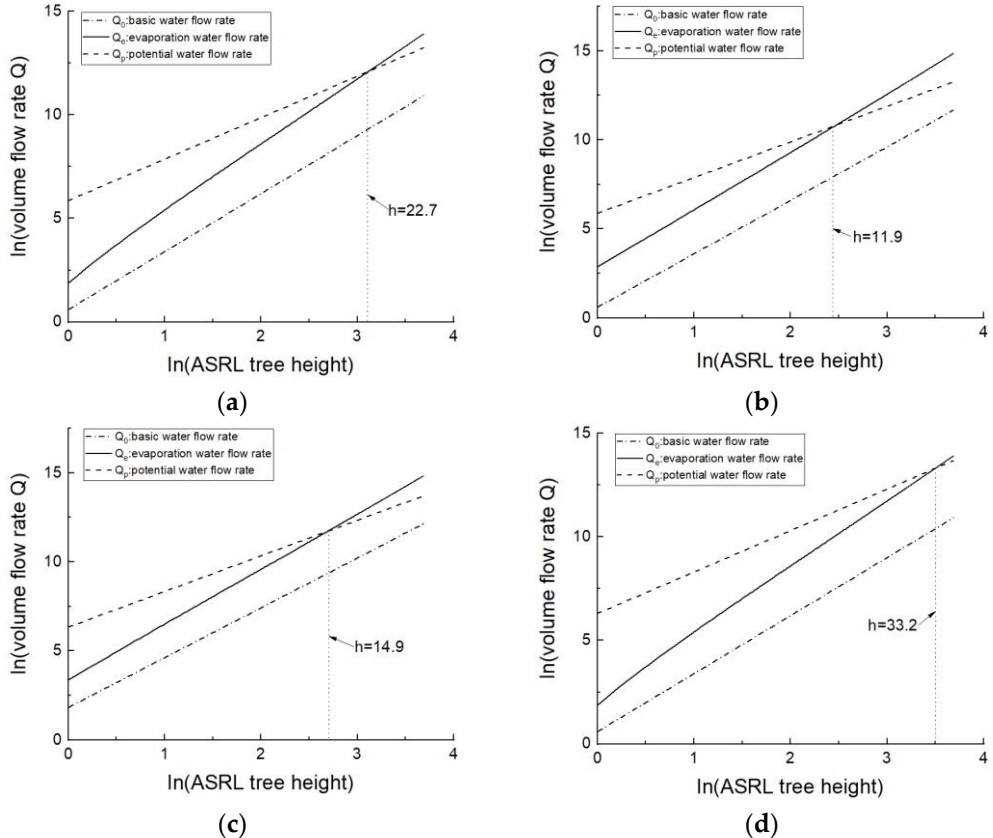

**Figure 7.** Analysis of optimization results. ASRL predictions of four case studies using verification point: (**a**) Model with parameters replacements, parametric optimizations, and topographic index; (**b**) Model with parametric optimizations and topographic index; (**c**) Model with parameters replacements and topographic; and (**d**) Model with parameters replacements and parametric optimizations. X axis represents the logarithm of tree height (unit: m), and Y axis represents the logarithm of water flow rate (unit: L/year).

*5.2. Model Sensitivity*

Sensitivity analysis presents the potential influence for predictions of the ASRL model by the climatic inputs, including precipitation, wind speed, vapor pressure, air temperature, and solar radiation. Changes in the water flow rates (Figure 8a–d) and maximum tree heights (Figure 8e–h) are investigated by perturbing each climatic variable while keeping the others constant. Intervals of variable alteration are 0.2 °C for temperature (ranging from −2 °C to 2 °C) and 2% for others (ranging from −20% to 20%). The monthly climatic variables of the verification point (121.554° E, 53.291° N) imported to the ASRL model are shown in Table 2.

As Figure 8 shows, the modeled water flow rates and potential maximum tree heights are sensitive to changes of climatic variables, and the direction and magnitude of model sensitivity are not the same across different variables. For instance, the potential water flow rate is influenced by precipitation, while the evaporation flow rate is sensitive to the others. A 20% increase in precipitation (Figure 8a) and vapor pressure (Figure 8e) produced a greater maximum tree height prediction ($\Delta h_{max} = 3.9$ m, $\Delta h_{max} = 10.9$ m). The modeled maximum tree heights are positively correlated with precipitation (Figure 8e) and vapor pressure (Figure 8f). In contrast, the predicted maximum tree height became smaller ($\Delta h_{max} = -3.4$ m, $\Delta h_{max} = -5.7$ m) when wind speed (Figure 8b) and air temperature (Figure 8d) were added, meaning a negative correlation between modeled tree height and the two variables. Comparing the slopes of the four curves in Figure 8e–h, the ASRL modeled maximum tree height is more sensitive to changes in vapor pressure (Figure 8g) and air temperature (Figure 8h) than

precipitation (Figure 8e) and wind speed (Figure 8f). Changes in wind speed and vapor pressure show contrary patterns of magnitude sensitivity.

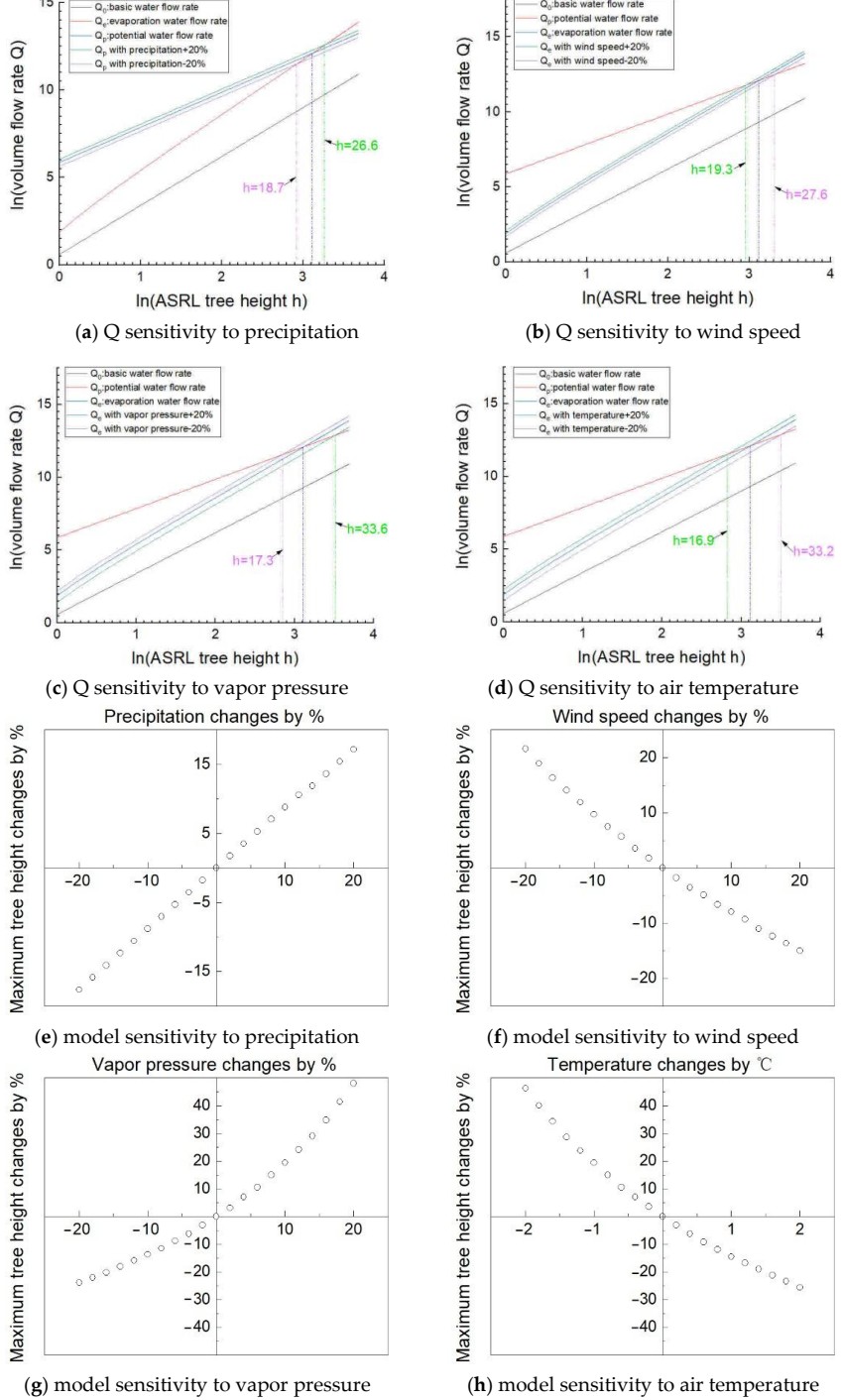

(**a**) Q sensitivity to precipitation

(**b**) Q sensitivity to wind speed

(**c**) Q sensitivity to vapor pressure

(**d**) Q sensitivity to air temperature

(**e**) model sensitivity to precipitation

(**f**) model sensitivity to wind speed

(**g**) model sensitivity to vapor pressure

(**h**) model sensitivity to air temperature

**Figure 8.** Sensitivity analysis of the ASRL model. The sensitivity to climatic variables including precipitation, wind speed, vapor pressure, and temperature. (**a**–**d**) Changes in the water flow rates are investigated by perturbing each climatic variable while keeping the others constant (precipitation, wind speed, and vapor pressure changed by ±20%, while temperature changed by ±2 °C). (**e**–**h**) Percent changes in maximum tree heights are investigated by perturbing each climatic variables while keeping others constant (precipitation, wind speed, and vapor pressure changed from −20% to 20% at a rate of 2%, while temperature changing from −2 °C to 2 °C at a rate of 0.2 °C).

**Table 2.** The monthly climatic inputs of the verification point.

| Group | prcp | wnd | vp | tmp | srad |
|---|---|---|---|---|---|
| January | 3 | 0.8 | 0.04 | −28.7 | 4.13 |
| February | 4 | 1 | 0.06 | −23.1 | 7.314 |
| March | 10 | 1.5 | 0.13 | −13.6 | 12.055 |
| April | 23 | 2.1 | 0.3 | −0.9 | 16.313 |
| May | 32 | 2.1 | 0.5 | 7.8 | 19.596 |
| June | 72 | 1.6 | 1.06 | 14.6 | 21.009 |
| July | 112 | 1.4 | 1.46 | 17.4 | 19.576 |
| Auguest | 102 | 1.3 | 1.28 | 14.8 | 16.104 |
| September | 49 | 1.5 | 0.68 | 7.1 | 11.953 |
| October | 19 | 1.5 | 0.29 | −4.2 | 8.073 |
| November | 9 | 1 | 0.11 | −19.1 | 4.649 |
| December | 5 | 0.7 | 0.05 | −27.4 | 3.116 |
| Unit | mm | m s$^{-1}$ | kPa | °C | MJ m$^{-2}$ day$^{-1}$ |

prcp, monthly total precipitation; wnd, mean wind speed; vp, mean vapor pressure; tmp, mean temperature; srad, mean solar radiation.

The ASRL model is least sensitive to solar radiation, similar to Choi's [27] result. For a 20% change in solar radiation, the predicted maximum tree height changes within 0.1 m. The reasons for this phenomenon are twofold: First, our research area is a water-limited environment, which means energy is not a major constraint on tree growth. Second, our study area belongs to a middle and high latitude region with low solar radiation. Due to the lack of experimental data, no more verification has been done.

## 6. Conclusions

In this paper, metabolic scaling theory and the Penman–Monteith equation are applied in the ASRL model to estimate maximum tree heights in the Greater Khingan Mountain, Inner Mongolia. Temperature, precipitation, wind speed, vapor pressure, and solar radiation are key input variables of the ASRL model. Model improvement and model sensitivity are also discussed to demonstrate the prognostic application of the ASRL model. Through our research, we found,

1. New values of the scaling coefficients $\phi$ and $\beta_3$ from field measurements make the model more consistent with the forest growth state of the study area.
2. Optimization of three parameters, $\beta_1$, $\gamma$, and $s_{leaf}$, improves the accuracy of the model prediction.
3. The introduction of a normalized topography index can effectively avoid overestimating short trees' heights in high slope areas and underestimating tall trees' heights in low slope areas.
4. Sensitivity analysis indicates the ASRL maximum tree height model is more sensitive to temperature and vapor pressure than any other climatic variables.

Caution is required in interpreting the results of the ASRL model because the current spatial scale fails to capture local tree height influenced by the small-scale climate variables, especially in mountains and valleys. Furthermore, species-specific parameters are not included in the model, which may also affect estimation of tree height and hence biomass. Thus, a progression of this work would be to account for application of high spatial resolution climate data and species information, and to assess the performance and utility of these techniques in other forests.

**Author Contributions:** Conceptualization, R.B.M., Y.S.; methodology, S.C., Y.S., Y.Z., X.N. and R.B.M.; investigation, Y.Z. and Y.S.; resources, X.N. and Y.Z.; writing-original draft preparation, Y.Z. and Y.S.; writing-review and editing, Y.Z. and Y.S.

**Funding:** This research was funded by National natural science foundation of China, grant number 41471312.

**Conflicts of Interest:** The authors declare no conflict of interest.

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
