# Peer review of "Mapping Maximum Tree Height of the Great Khingan Mountain, Inner Mongolia Using the Allometric Scaling and Resource Limitations Model"

_forests, doi:10.3390/f10050380_

Round 1

Reviewer 1 Report

The paper present a well designed improvement of an existing model (ASRL model). The authors achieved to map precisely maximumtree height of the Great Khingan Mountain, in Mongolia.

Though relatively short, the paper is well written. Some parts would therefore need to be developped. Indeed, some results are presented in the discussion section. These results should be placed in the appropriate section (Results one), and should be discussed a little more in the discussion part.

The authors will find some detailed comments in the file attached (comments in the pdf version of the manuscript).

I hope my comments will improve the quality of the manuscript.

Author Response

Dear reviewer,

I am grateful to you for your valuable insights and comments. All the comments have been carefully read, and I have responded to them.

Besides, please accept my apologies for my oversight that i have not add the line number and page number in my manuscript. 

With best wishes,

Yao Zhang

Reviewer 2 Report

The authors propose an improved ASRL model to estimate forest canopy height by replacing some parameters of the ASRL model, with optimal ones derived using field-data acquired over the study area in Mongolia (with the help of metabolic scaling theory and PM equations). At this point the paper is not in a good shape and the authors needs to address several issues as stated in the comments below. Also, the English used is very poor, which makes it very difficult to clearly understand the paper.

Author Response

Dear Reviewer,

I am grateful for your valuable insights and comments to improve the manuscript. All the comments have been carefully read, and I have responded to them as suggested by the reviewers.

Besides, my apologies for my carelessness. I have added the line number and page number in my latest manuscript.

With best wishes!

Yao Zhang 

Round 2

Reviewer 2 Report

The authors have addressed all the concerns in the revised version of the thesis.